# Genome-Wide Analysis of Specific *PfR2R3-MYB* Genes Related to Paulownia Witches’ Broom

**DOI:** 10.3390/genes14010007

**Published:** 2022-12-20

**Authors:** Xiaogai Zhao, Bingbing Li, Xiaoqiao Zhai, Haifang Liu, Minjie Deng, Guoqiang Fan

**Affiliations:** 1Institute of Paulownia, Henan Agricultural University, Zhengzhou 450002, China; 2Forestry Academy of Henan, Zhengzhou 450002, China; 3College of Forestry, Henan Agricultural University, 95 Wenhua Road, Jinshui District, Zhengzhou 450002, China

**Keywords:** *Paulownia fortunei*, *R2R3-MYB* family, *PfR2R3-MYB15*, branching, subcellular localization, yeast two-hybrid, bimolecular fluorescence complementation

## Abstract

Paulownia witches’ broom (PaWB), caused by phytoplasmas, is the most devastating infectious disease of Paulownia. *R2R3-MYB* transcription factors (TF) have been reported to be involved in the plant’s response to infections caused by these pathogens, but a comprehensive study of the *R2R3-MYB* genes in Paulownia has not been reported. In this study, we identified 138 *R2R3-MYB* genes distributed on 20 chromosomes of *Paulownia fortunei*. These genes were classified into 27 subfamilies based on their gene structures and phylogenetic relationships, which indicated that they have various evolutionary relationships and have undergone rich segmental replication events. We determined the expression patterns of the 138 *R2R3-MYB* genes of *P. fortunei* by analyzing the RNA sequencing data and found that *PfR2R3-MYB15* was significantly up-regulated in *P. fortunei* in response to phytoplasma infections. *PfR2R3-MYB15* was cloned and overexpressed in *Populus trichocarpa*. The results show that its overexpression induced branching symptoms. Subsequently, the subcellular localization results showed that *PfR2R3-MYB15* was located in the nucleus. Yeast two-hybrid and bimolecular fluorescence complementation experiments showed that *PfR2R3-MYB15* interacted with *PfTAB2*. The analysis of the *PfR2R3-MYB15* gene showed that it not only played an important role in plant branching, but also might participate in the biosynthesis of photosystem elements. Our results will provide a foundation for future studies of the *R2R3-MYB* TF family in Paulownia and other plants.

## 1. Introduction

Transcription factors (TFs) are DNA-binding proteins that inhibit or activate the transcription of their target genes and form complex regulatory networks [1]. The MYB TFs comprise a large TF family and play important roles in the life cycle of plants [2]. In 1982, MYBs were first discovered in the avian leukosis virus [3], and subsequently were found in *Zea mays* (corn) [4], *Arabidopsis thaliana* (*A. thaliana*) [5], *Gossypium* (cotton) [6], *Glycine max* (soybean) [7], *Populus trichocarpa* (populus) [8] and *Ziziphus jujube* (Chinese jujube) [9]. MYB’s domain commonly includes one to four incomplete repeats (R), which consist of 50–53 conserved amino acid residues [10]. Based on the number of R repeats, the MYB proteins can be divided into four types, including 1R (R1, R2 and R3-MYB), 2R (R2R3-MYB), 3R (R1R2R3-MYB) and 4R (R1R2R2R1 and R1R2R2R2-MYB) proteins, of which the R2R3-MYB is the most common type in plants [10].

*R2R3-MYB*s regulate morphogenesis [10], growth and development [11], hormone response [12,13], and resistance to stress [14]. *R2R3-MYB* genes have been identified and analyzed in many plants. In *A. thaliana*, *AtMYB59* was shown to negatively regulate cell cycle and root growth. Compared to the wild type, the overexpression of *AtMYB59* produced shorter and fewer roots, and the deletion mutant *AtMYB59-1* had longer roots [15]. *AtMYB5* and *AtMYB123* are involved in tannin biosynthesis [16]. The overexpression of *AtMYB30* in *A. thaliana* enhanced its hypersensitive response (HR) and tolerance to pathogen attacks [17]. Further research found that resistance against different bacterial pathogens and HR cell death are strongly decreased in antisense *AtMYB30* lines [17]. Furthermore, *AtMYB96* enhanced the resistance to drought stress by integrating auxin and abscisic acid signals in *A. thaliana* [18], and *AtMYB38* was found to be involved mainly in the formation of axillary meristem during plant branch development [19]. Many studies have shown that *R2R3-MYB*s play essential roles in corn, *A. thaliana*, rice, cotton and poplar, but their roles in Paulownia have not been reported so far.

*Paulownia fortunei* (*P. fortunei*) is a fast-growing tree species that has been planted worldwide for its high economic and ecological value [4,20]. Paulownia witches’ broom (PaWB), which is caused by phytoplasma belonging to the aster yellows’ group ‘Candidatus Phytoplasma asteris’, can result in great losses in the forest industry [21]. Witches’ brooms and the yellowing of leaves are the main symptom of PaWB [22]. In order to study the role of the MYB family in the pathogenesis of PaWB, we identified 138 *R2R3-MYB* TFs in *P. fortunei* and analyzed their structure, functions, phylogenetic relationships, and the expression patterns of the *R2R3-MYB* genes of PaWB. We found that *PfR2R3-MYB15* was significantly up-regulated in *P. fortunei* in response to phytoplasma infestations by analyzing RNA sequencing (RNA-Seq) data. To study the relationship between *PfR2R3-MYB15* and PaWB, transgene, subcellular localization, yeast two-hybrid (Y2H) screening, and bimolecular fluorescence complementation (BiFC) were used to identify the function of *PfR2R3-MYB15*. Our results will provide a basis for future studies of the *R2R3-MYB* family and contribute toward our understanding of the important role of *PfR2R3-MYB15* in plant branching.

## 2. Materials and Methods

### 2.1. Plant Materials

The tissue-cultured seedlings used in this study were from the Institute of Paulownia, Henan Agricultural University, Zhengzhou, Henan Province, China. Healthy *P. fortunei* seedlings (PF), phytoplasma-infected *P. fortunei* seedlings (PFI), and wild-type *P. trichocarpa* (WT) were cultured as described by Zhai et al. (2010) [22]. PF and PFI were collected at 30 days, immediately frozen in liquid nitrogen, and stored at −80 °C for further study.

### 2.2. Identification PfR2R3-MYB Genes in P. fortunei

The *P. fortunei* genome data were obtained from the NCBI genome database (https://www.nc-bi.nlm.nih.gov, accessed on 16 June 2021). *PfR2R3-MYB* genes were identified in accordance with Jin et al. (1999) [23]. *R2R3-MYB* genes in the *P. fortunei* genome were respectively identified using the Hidden Markov model (HMM) file of the MYB DNA-binding domain (PF00249) [24] and a local BLAST analysis (based on the *A. thaliana R2R3-MYB* gene sequences) (https://www.arabidopsis.org, accessed on 16 June 2021). The results from the two methods were compared to identify *PfR2R3-MYB* genes. The Conserved Domain Database (https://www.ncbi.nl-m.nih.gov/Structure/bwrpsb/bwrpsb.cgi, accessed on 18 June 2021) and Pfam were used to confirm the *PfR2R3-MYBs*. Then, the physical and chemical properties of *PfR2R3-MYB* proteins were analyzed using ExPASy tools (https://web.expasy.org/protparam/, accessed on 18 June 2021). Plant-mPLoc (http://www.csbio.sjtu.edu.cn/bioinf/plantmulti/, accessed on 18 June 2021) was used to predict the subcellular localization of the *PfR2R3-MYB* proteins.

### 2.3. Chromosomal Location and Collinear Evolution Analysis of PfR2R3-MYB Genes

Chromosome locations and lengths of the *PfR2R3-MYB* genes were obtained using graphics of TBtools software (https://github.com/CJ-Chen/TBtools, accessed on 20 June 2021) [25]. Subsequently, the genes were named on the basis of their position and order on the chromosomes. Then, fragment repetition events and collinearity relationships of the *R2R3-MYB* genes in *P. fortunei* and *A. thaliana* were determined using dual synteny plot on TBtools platform (https://github.com/CJ-Chen/TBtools, accessed on 20 June 2021). Non-synonymous/synonymous substitution ratios (Ka/Ks) of *PfR2R3-MYB* homologous gene pairs were calculated using MCScan on TBtools (https://github.com/CJ-Chen/TBtools, accessed on 20 June 2021) [25,26].

### 2.4. Phylogenetic and Conserved Motif Domain Analysis of PfR2R3-MYB Genes

To determine the phylogenetic relationships among the *PfR2R3-MYB* proteins, we compared their amino acid sequences (https://www.ncbi.nlm.nih.gov, accessed on 22 June 2021) with those of *A. thaliana* (https://www.arabidopsis.org/, accessed on 22 June 2021). A phylogenetic tree was constructed using the neighbor-joining (NJ) adjacency method in MEGA7.0 described by Dubos et al. (2006). NCBI CD-Search (https://www.ncbi.nlm.nih.gov/Structure/cdd/wrpsb.cgi, accessed on 22 June 2021) described by Zhou et al. (2006) was used to determine the conserved motifs of the *PfR2R3-MYB*s.

### 2.5. Gene Structure and Promoter Cis-Acting Element Analysis of PfR2R3-MYB Genes

The exon and intron structures for individual *PfR2R3-MYB* genes were checked by aligning the cDNA sequences to the corresponding genomic DNA sequences [26]. The gene structures of the *PfR2R3-MYBs* were visualized using the online software GSDS (http://gsds.cbi.pku.edu.cn/ accessed on 18 June 2021). Plant CARE (http://bioinformatics.psb.ugent.be/webtoo-ls/plantc-are/html/, accessed on 24 June 2021) was used to predict promoter cis-acting element [27].

### 2.6. Expression Pattern Analysis of PfR2R3-MYB Genes Responsive to Phytoplasma and qRT-PCR Verification

In previous studies, our research group found that the phytoplasma content decreased in PFI treated with 100 mg·L^−1^ of rifampin (Rif) with the extension in treatment time, and the plant morphologically showed reduction in branching and gradually reached a healthy state [28]. To further understand the role of *PfR2R3-MYB* genes in the occurrence of PaWB, we downloaded the RNA-seq data (https://www.ncbi.nlm.nih.gov/sra/?term=Paulownia, accessed on 27 June 2021), including PF and PFI, as well as PFI treated with 100 mg·L^−1^ Rif for 5, 10 and 20 days (PFI100-5, PFI100-10, and PFI100-20) (SRR11787883, SRR11787894, SRR11787905, and SRR11787912, and SRR11787921). Clean reads were aligned to the *P. fortunei* genome using BWA software. Gene expression levels were quantified by RSEM (RNASeq by expectation maximization) software package. The fragments per kb per million reads (FPKM) values were used to calculate the expression of genes. The heat map was generated using heatmap on TBtools. To validate the sequencing data, qRT-PCRs were performed to detect the expression patterns of eight differentially expressed genes (DEGs). We designed mRNA and universal reverse primer using Primer5 (Appendix A). The qRT-PCRs were performed on a CFX 96 Touch RT-PCR detection system (Bio-Rad, Hercules, CA, USA) with SYBR Green Real-time PCR Master Mix (Toyobo, Osaka, Japan) as described previously [29]. Expression levels were calculated by the 2^−ΔΔCT^ method [30]. Actin (Forward primer: AATGGAATCTGCTGGAAT, Reverse primer: ACTGAGGACAATGTTACC) was used as the internal reference gene. Three biological replicates were used.

### 2.7. Vector Construction and Agrobacterium Tumefaciens Transformation

The *PfR2R3-MYB15* gene primers were as follows: forward primer was 5′-ATGGGAA GAGCACCTTGCTG-3′ and reverse primer was 5′-CAAATATCAACAGAAAGTTCTGAGTTTC-3′. The *PfR2R3-MYB15* gene was driven by the cauliflower mosaic virus (CaMV) 35S promoter and was transformed into *Agrobacterium tumefaciens* (*A. tumefaciens*) by the freeze–thaw method [31]. WT was cultured on 1/2 woody plant medium (WPM) for 30 days, then stems were infected with *A. tumefaciens* GV3101 harboring the *PfR2R3-MYB15* gene using cultures with an OD at 600 nm of 0.6; the details of this method referred to Song et al. (2006).

### 2.8. Detection of Transgenic Seedlings

PCR was used to identify the transgenic seedlings. The primer sequences designed by Primer Premier 5.0 were as follows: forward primer was 5′-GGATCCATGGGAAGAGCACC-3′ and reverse primer was 5′-GAGCTCCTATTTGTACAATT-3′. The PCR cycling conditions were 95 °C for 3 min followed by 35 cycles of 95 °C for 30 s, 52 °C for 30 s, and 72 °C for 90 s, and melting at 72 °C for 10 min. QRT-PCR was used to validate *OE-PfR2R3-MYB15*. The method is the same as that in Section 2.6. Ribosomal *PtActin* RNA was used as the reference gene. The primer sequences were designed by Primer Premier 5.0 (Appendix A).

### 2.9. Subcellular Localization of PfR2R3-MYB15 Protein

To verify the subcellular localization of the *PfR2R3-MYB15* protein, we constructed pBI121 vectors carrying 35S::CDS *PfR2R3-MYB15*-GFP and 35S::GFP. The primer sequences were designed and are listed in Appendix A. The 35S::*PfR2R3-MYB15-GFP* and 35S::*GFP* were transfected into *Nicotiana benthamiana* (*N. benthamiana*) cells, as described by Xie et al. (2020).

### 2.10. Yeast Two-Hybrid Assays

The coding region of *PfR2R3-MYB15* was cloned into pGBKT7 as the bait. Toxic and auto-activation tests were performed. PFI cDNA libraries were screened using pGBKT7–*PfR2R3-MYB15* prey plasmids [28]. The coding regions of *Paulownia_LG3G001014* (protein TAB2 homolog, *PfATAB2*) were cloned into pGADT7 (Appendix A). PGADT7-largeT/pGBKT7-laminC (negative control), pGADT7-largeT/pGBKT7-p53 (positive control) and *PfR2R3-MYB15* (pGBKT7)/*PfTAB2* were transfected into AH109 using a Yeast-Maker yeast transformation system. Plasmid isolation, yeast transformation, mating, and interaction test were performed using a Matchmaker GAL4TM Two-Hybrid System 3 in accordance with the manufacturer’s instructions.

### 2.11. Bimolecular Fluorescence Complementation Assays

Bimolecular fluorescence complementation (BiFC) assays were performed by constructing pNC-ENN-*PfR2R3-MYB15* and pNC-ECN-*PfATAB2* using a Nimble cloning system. Vector transformation, *N. benthamiana* injection, and fluorescent observation were conducted as described by Xin et al. (2020).

### 2.12. Data Availability Statement

All the raw read data will be provided during review.

## 3. Results

### 3.1. Genome-Wide Identification of PfR2R3-MYB in P. fortunei

We identified 245 MYB genes in the *P. fortunei* genome; 101 were *R1-MYBs*, 6 were *R1R2R3-MYBs* and 138 were *R2R3-MYBs* (Appendix A). On the basis of their chromosome localization, we named the 138 *PfR2R3-MYB* genes from *PfR2R3-MYB1* to *PfR2R3-MYB138* (Appendix A). The length of *PfR2R3-MYB* CDSs varied from 582 bp (*PfR2R3-MYB41*) to 3516 bp (*PfR2R3-MYB115*), and the size of the proteins ranged from 193 (*PfR2R3-MYB41*) to 1171 (*PfR2R3-MYB115*) amino acids (Appendix A). The isoelectric points (PIs) of the proteins ranged from 4.75 (*PfR2R3-MYB29*) to 9.58 (*PfR2R3-MYB83*), and the relative molecular mass was from 22,252.21 (*PfR2R3-MYB41*) to 129,740.84 D (*PfR2R3-MYB115*) (Appendix A). The *PfR2R3-MYB* proteins were predicted to be located in the cell nucleus, which is consistent with the location of *R2R3-MYB* proteins in *Arabidopsis* [2] and poplar [8].

### 3.2. Localization, Gene Replication and Collinearity Analysis

The 138 *PfR2R3-MYB* genes were assigned to the 20 chromosomes of *P. fortunei* (Figure 1a), and their distribution on the chromosomes was scattered and widespread, which was consistent with the distribution of the *PtR2R3-MYB* genes on the poplar chromosomes [8]. Sixteen *PfR2R3-MYB*s were on chromosome 6, eleven each on chromosomes 11 and 13, six each on chromosomes 4, 8, 12, 14, 16, and 20, five each on chromosomes 10, 15, 17, and 19, four each on chromosomes 5 and 8, and three on chromosome 1.

Gene duplication is important for gene family formation and extension in plants [3]. In this study, we performed a collinearity analysis of the *PfR2R3-MYB* genes and identified 92 *PfR2R3-MYB* repeat pairs (Figure 1b). The *PfR2R3-MYB* gene duplication events were fragment duplications in all the *P. fortunei* chromosomes, implying that 92 fragment duplications played important roles in the evolution of the *PfR2R3-MYB* gene family (Appendix A). To assess the limiting factors of the evolution of the *PfR2R3-MYB* gene family, we calculated the Ka/Ks ratios of the 92 repeat pairs and found that they ranged from 0.071 (*PfR2R3-MYB76*/*PfR2R3-MYB86*) to 0.546 (*PfR2R3-MYB58*/*PfR2R3-MYB103*) (Appendix A). The ratio was much less than 1, which indicated that the *PfR2R3-MYB* gene family had undergone gene purification and selection in the evolutionary process.

To further investigate the evolution of the *R2R3-MYB* gene family in *P. fortunei* and *A. thaliana*, we constructed a collinearity map of 138 *PfR2R3-MYBs* and 126 *AtR2R3-MYBs*, which represented 182 *PfR2R3-MYB* orthologous gene pairs (Appendix A). The results showed that the *PfR2R3-MYBs* had common ancestors and were highly conserved during the evolutionary process.

### 3.3. Analysis of Chromosomal Location and Synteny of PfR2R3-MYB Genes

To clarify the classification of *PfR2R3-MYB* proteins, we constructed an NJ rootless phylogenetic tree using 138 *PfR2R3-MYB* and 126 *AtR2R3-MYB* protein sequences from *P. fortunei* and *A. thaliana* (Figure 2). The 138 *PfR2R3-MYB* proteins were divided into 27 subfamilies (named I–XXVII), most of which were consistent with those of *A. thaliana* in Figure 2. However, none of the *PfR2R3-MYB* proteins were found in subfamily XVII, suggesting that some *R2R3-MYB* genes may have been lost during the long-term evolution of the *P. fortunei* genome. Subfamilies I and XXIII each had twelve *PfR2R3-MYB*s, accounting for 8.63% of the total *P. fortunei PfR2R3-MYB* family, whereas subfamily XXVII had only one *PfR2R3-MYB*. Based on the evolutionary relationship of *R2R3-MYB* proteins in *P. fortunei* and poplar, we found that the *138R2R3-MYB* and 192 MYB proteins mostly clustered in a branch, showing a sister-group relationship (Appendix A).

### 3.4. Conserved Domains and Gene Structure Analyses of PfR2R3-MYBs

We identified 10 conserved motifs in the 138 *PfR2R3-MYB*s (Figure 3b, Appendix A). The motifs were widely distributed in the *PfR2R3-MYB*s. Motifs 1, 2, 3, 4, 5, 8 and 10 were detected in *PfR2R3-MYB128, 136, 133, 127, 103, 138* and *117*, respectively, and motifs 6, 7 and 9 were detected in *PfR2R3-MYB59, 30* and *27*, respectively. Motifs 4, 6 and 9 had one highly conserved tryptophan site, motifs 1 and 10 had two highly conserved tryptophan sites, and motifs 3 and 5 had three highly conserved tryptophan sites. The conserved motifs are consistent with those found in poplar *R2R3-MYB*s [8].

The intron and exon positions of the *PfR2R3-MYB* genes were visualized using the GSDS software. Significant differences and diversity were observed in the *PfR2R3-MYB* genes’ structure (Figure 3c). Eight *PfR2R3-MYBs* had no introns (*PfR2R3-MYB2*, *PfR2R3-MYB18*, *PfR2R3-MYB43*, *PfR2R3-MYB81*, *PfR2R3-MYB82*, *PfR2R3-MYB98*, *PfR2R3-MYB113*, *PfR2R3-MYB137*), whereas most of the *PfR2R3-MYBs* had three exons and two introns, and *PfR2R3-MYB46* had the highest number of introns (11) and exons (12). This result indicated that the functions of some *PfR2R3-MYBs* might change during the evolution of the *PfR2R3-MYB* family.

### 3.5. Prediction of Cis-Regulatory Elements in PfR2R3-MYB Gene Family

To better understand the potential functions of the *PfR2R3-MYB*s, we detected *cis*-regulatory elements in the 2-kb upstream sequences of the *PfR2R3-MYB* genes using the PLACE database. We found that most of the *cis*-acting elements were light-responsive elements, whereas others were defense- and stress-responsive as well as hormone-responsive elements. Some common *cis*-acting elements were also found in the promoter regions of the *PfR2R3-MYB* genes (Figure 4, Appendix A). The regulation of gene expression is mainly dependent on the presence of *cis*-regulatory elements in the gene promoter regions [32]. The presence of light-responsive, stress-responsive and hormone-responsive elements in the upstream sequences leads us to speculate that they may have important roles in the *PfR2R3-MYB* gene family.

### 3.6. PfR2R3-MYBs Involved in the Phytoplasma Interaction and Verification of qRT-PCR

We analyzed the expression patterns of the 138 *PfR2R3-MYBs* in phytoplasma-infected *P. fortunei* via RNA-Seq. The results showed that 119 *PfR2R3-MYBs* responded to phytoplasma infestation (Figure 5a) and 26 (15 down-regulated and 11 up-regulated) of them were significantly differentially expressed in PF versus PFI (Appendix A), indicating that these genes may actively respond to the PaWB phytoplasma. In previous studies, our research group found that the phytoplasma content decreased in PFI treated with 100 mg·L^−1^ of Rif with an extension of the treatment time, and the plant morphologically showed a reduction in branching and gradually reached a healthy state [28]. The 26 *PfR2R3-MYBs* actively responded to the PaWB phytoplasma in Rif 100-5/PFI, Rif100-10/PFI and Rif100-20/PFI (Figure 5a).

To validate the accuracy of the RNA-Seq results, we randomly selected eight *PfR2R3-MYB* differentially expressed genes in PFI vs. PF for the qRT-PCR analysis. The trends of their expression were similar, which indicated that the RNA-Seq data were reliable (Figure 5b–i).

### 3.7. Morphological Change of P. trichocarpa Overexpressing PfR2R3-MYB15

In order to analyze the function of *PfR2R3-MYB15*, *PfR2R3-MYB15* was cloned and overexpressed in *P. trichocarpa* via a agrobacterium-mediated method. To confirm that *PfR2R3-MYB15* was integrated into the genome of *P. trichocarpa*, we used three traditional methods to identify three over-expressed strains (*OE-PfR2R3-MYB15-1*, *OE-PfR2R3-MYB15-2* and *OE-PfR2R3-MYB15-*3). The root morphology, PCR and qRT-PCR methods were used to detect *OE-PfR2R3-MYB15* (Figure 6a,b). The results showed that *OE-PfR2R3-MYB15-1, OE-PfR2R3-MYB15-2* and *OE-PfR2R3-MYB15-3* were positive.

To further study the functions of *PfR2R3-MYB15,* the positive seedlings were transplanted into pots. When the plants were cultured up to 90 days, the phenotypic characteristics showed that *OE-PfR2R3-MYB15* had obvious branches compared with WT (Figure 6). The results show that *PfR2R3-MYB15* plays an important role in plant branching symptoms, which lays a foundation for future studies of the *R2R3-MYB* family in other plants.

### 3.8. PfR2R3-MYB15 Located in the Nucleus

To verify the predicted nucleus location of the *PfR2R3-MYB15* protein, 35S::*PfR2R3-MYB15-GFP* was transfected into *N. benthamiana* cells. The fluorescence signal of MYB15-GFP was detected in the nucleus, whereas GFP (the positive control) was in the full cell. The results confirmed that *PfR2R3-MYB15* was located in the nucleus (Figure 7), which was consistent with the Plant-mPLoc prediction.

### 3.9. The Discovery of Related Proteins with PfR2R3-MYB15 as Bait

*PfR2R3-MYB15* self-activating activity was detected by Y2H screening. The results showed that 30 mM 3-AT can inhibit the autoactivation of *PfR2R3-MYB15* (Figure 8a,c). To explore the molecular mechanism of *PfR2R3-MYB15*, a Y2H screen was performed to identify the potential binding partners of the *PfR2R3-MYB15* protein. A total of 24 proteins from the cDNA library of PFI were found to interact with *PfR2R3-MYB15* (Appendix A). Among them, it has been verified by Y2H screening and BiFC that *PFR2R3-MYB15* interacted with *PfTAB2* (Paulownia_LG3G001014) (Figure 8c). *ATAB2* was involved in the biogenesis of photosystem I (PSI) and II (PSII) [33]. We speculate that *PfR2R3-MYB15* may be involved in the biosynthesis of the photosystem elements, but this needs to be further verified in future studies.

## 4. Discussion

*R2R3-MYBs* play important roles in plant growth and development, morphogenesis and resistance to stress [11,12,13]. The *R2R3-MYB* family has been widely studied in plants. In *A. thaliana*, 126 *R2R3-MYBs* in 25 subfamilies have been identified and their functions have been predicted by phylogenetic and RNA-Seq analyses [2,34]. In maize, 157 *R2R3-MYB* TFs with diverse expression patterns and different functions have been reported [35]. In *Solanum lycopersicum*, 121 *R2R3-MYB* TFs were found, and the RNA interference of pGSMyb12 was shown to decrease the lateral meristems [36]. Despite their important roles in other plants, this study is the first to report *R2R3-MYBs* in the *P. fortunei* genome and analyze their physicochemical properties, structure and evolution. Furthermore, we analyzed the expression patterns of the *R2R3-MYB* genes in PF and PFI and focused on the functional validation of the significantly overexpressed gene *PfR2R3-MYB15*. Our results provide new perspectives on the roles of *PfR2R3-MYB* TFs in plants’ responses to PaWB infestations.

We identified 138 *PfR2R3-MYB* genes in the *P. fortunei* genome (Figure 1), which is more than those in *A. thaliana* [5], but less than those in *Populus* [8]. We speculated that the number of gene family members may be related to the genome size (Figure 1 and Figure 2). Phylogenetic trees based on the *R2R3-MYB* protein sequences of *P. fortunei* and *A. thaliana* showed that the 138 *PfR2R3-MYB* genes clustered into 27 subfamilies (named I–XXVII), most of which contained *A. thaliana R2R3-MYB* TFs. Some exceptions were the XXI subfamily, which did not have any *AtR2R3-MYBs*, and the XVII subfamily, which did not have any *PfR2R3-MYBs*, implying that the *R2R3-MYB* gene family had undergone great differentiation in the evolution of these species. The most closely related genes likely have similar functions. For example, in *A. thaliana*, the overexpression of *AtRAX2* resulted in an increased number of branches compared with the wild type, whereas the number of branches decreased in the *AtRAX2* mutant. In the XXIII subfamily, *PfR2R3-MYB15* was most closely related to At2g36890 (RAX2), suggesting that *PfR2R3-MYB15* may have similar functions to AtRAX2.

Plant genomes are known to undergo whole-genome replication, segmental replication and tandem repetition events which contribute to the expansion of distinct gene families. Gene duplication involves three mechanisms: tandem replication, segmental duplication and transposition events [37]. Fragment and tandem duplication are the main evolutionary mechanisms for gene family expansion and evolution in plants [37]. In this study, the chromosome distribution analysis showed that the *PfR2R3-MYBs* were unevenly distributed on the 20 chromosomes. The collinearity analysis indicated that 92 fragment duplication events had occurred. These results suggested that fragment duplication played an important role in the evolution of the *PfR2R3-MYB* gene family. We also found that the Ka/Ks ratios of the *P. fortunei* segmental repeat pairs were much less than one, implying that the *PfR2R3-MYB* family experienced strong purification selection during its evolution. The collinear relationship of the *R2R3-MYB* genes in *A. thaliana* and *P. fortunei* showed that 105 *PfR2R3-MYBs* and 99 *AtR2R3-MYBs* formed 182 lineal homologous gene pairs (Figure 1c). AtRAX1 (MYB37) proteins interact with CUC2 to further regulate axillary meristems’ expression and control lateral bud production [38]. RAX1, RAX2 (AtMYB38) and RAX3 (AtMYB84) have obvious effects on branch development in *A. thaliana*. Plants overexpressing *RAX2* showed lateral bud symptoms, whereas *RAX2* knockout mutants had fewer lateral buds than the wild type. Furthermore, the knockout line *RAX1/RAX2/RAX3* was shown to have significantly reduced branches [19]. Together, our results provide new ideas for studying the evolutionary relationships of the *PfR2R3-MYB* gene family in *P. fortunei*.

Numerous studies have shown that members of the *PfR2R3-MYB* family play important roles in plant branching, so we analyzed the expression patterns of the *PfR2R3-MYB* genes in PF and PFI by RNA-Seq. We detected 119 *PfR2R3-MYB* genes that responded to PaWB phytoplasma infestationS, and 26 (15 down-regulated and 11 up-regulated) of them were significantly differentially expressed in PF versus PFI (FC ≥ 2.0, false discovery rate < 0.01) (Figure 5a). We focused on *PfR2R3-MYB15*, which was significantly up-regulated in PF versus PFI. To confirm its function, *PfR2R3-MYB15* was cloned and transfected into *P. trichocarpa*, and its overexpression induced lateral bud symptoms (Figure 6b). Y2H and BiFC experiments demonstrated that *PfR2R3-MYB15* interacted with *PfTAB2* (Paulownia_LG3G001014). Barneche et al. (2006) found *PfTAB2* was the localization of the chloroplast. It may be due to the localization of *PfTAB2* that *PfR2R3-MYB15* interacted with *PfTAB2* in the cell membrane and nucleus via BiFC. Previous studies have shown that *ATAB2* is involved in the biogenesis of photosystems I (PSI) and II (PSII) among the *PfRAX2*-related proteins that are involved in photosynthesis [33]. In this study, we speculate that *PfR2R3-MYB15* might participate in the biosynthesis of the photosystem by influencing *PfTAB2*. These results provide a valuable resource to explore the potential functions of *PfR2R3-MYB* genes and provide a foundation for future studies of *the R2R3-MYB* TF family in Paulownia and other plants.

## 5. Conclusions

In this study, we identified 138 *P. fortunei R2R3-MYB* TFs, which were distributed on 20 chromosomes and were divided into 27 groups. Their gene structure, phylogenetic relationship, *cis*-regulatory elements, evolutionary relationships and rich segmental replication events were analyzed. By combining RNA-seq data, we profiled the expression patterns of the 138 *PfR2R3-MYB* members in phytoplasma-infected seedlings of *P. fortunei*. We found that *PfR2R3-MYB15* is significantly up-regulated in infected *P. fortunei*. The overexpression of *PfR2R3-MYB15* induced branching symptoms in *P. trichocarpa*. The yeast two-hybrid screening and bimolecular fluorescence complementation showed that *PfR2R3-MYB15* interacted with *PfTAB2*. Our results will provide a foundation for future studies of the *R2R3-MYB* TF family in Paulownia.

## Figures and Tables

**Figure 1 genes-14-00007-f001:**
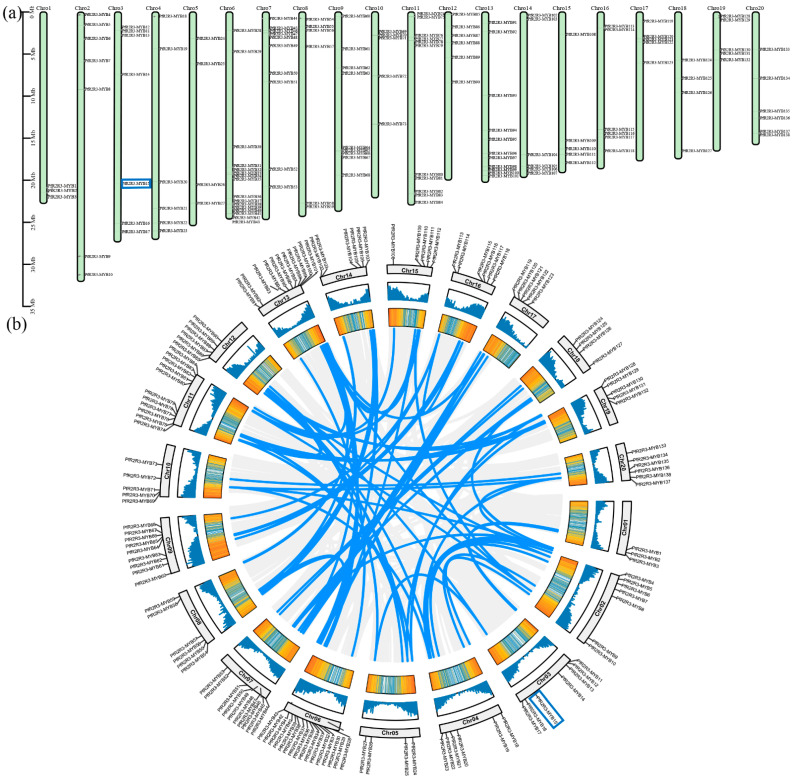
Localization and collinearity analysis of *PfR2R3-MYBs*. (**a**) 138 *PfR2R3-MYB* genes were randomly distributed on *P. fortunei* chromosomes. (**b**) The duplicated *PfR2R3-MYB* gene relationship. The blue line represents the gene pair of *PfR2R3-MYB*.

**Figure 2 genes-14-00007-f002:**
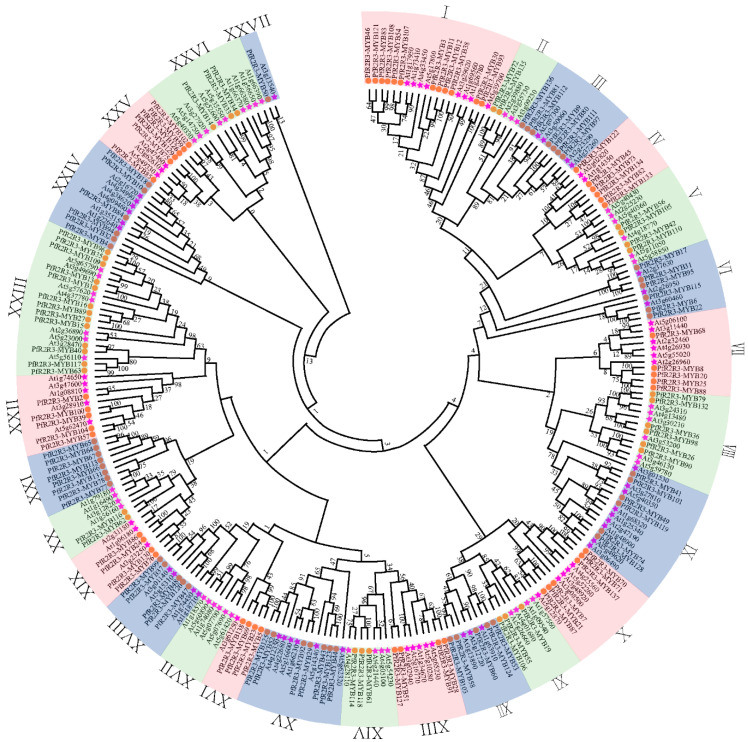
Phylogenetic analysis of the *R2R3-MYB*s proteins in *P. fortunei* and *A. thaliana*. Phylogenetic tree of *R2R3-MYB* family genes in *P. fortunei* and *A. thaliana*. MEGA 7.0 was used to generate the unrooted neighbor-joining tree with 1000 bootstrap replicates. The *R2R3-MYB*s from *P. fortunei* and *A. thaliana* were divided into 27 groups. The *R2R3-MYB* genes are labeled in yellow and pink for *P. fortunei* and *A. thaliana*, respectively.

**Figure 3 genes-14-00007-f003:**
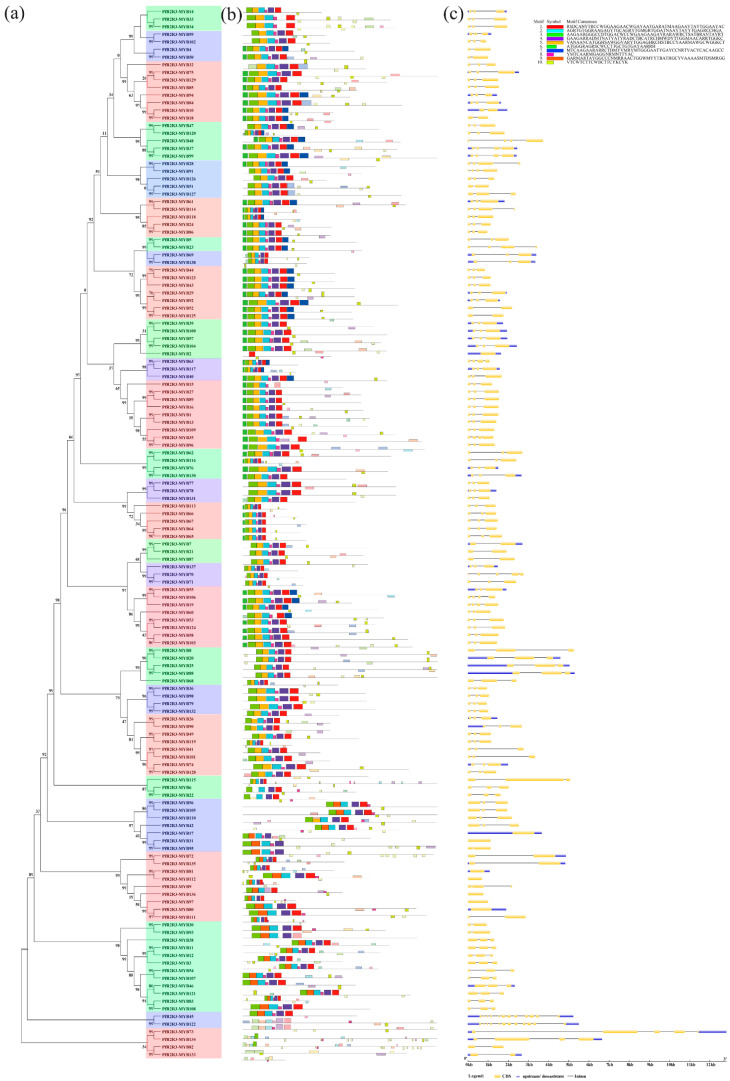
Phylogenetic tree, conserved motifs and gene structures of *R2R3-MYB* family members. (**a**) Phylogenetic tree was constructed based on 138 *R2R3-MYB* protein sequences, which were divided into 27 groups, according to phylogenetic tree. (**b**) Architecture of conserved protein motifs in 138 *PfR2R3-MYB* members. Each motif is represented by the number on the colored box. (**c**) Exon/intron structures of *R2R3-MYB* genes from *P. fortunei*.

**Figure 4 genes-14-00007-f004:**
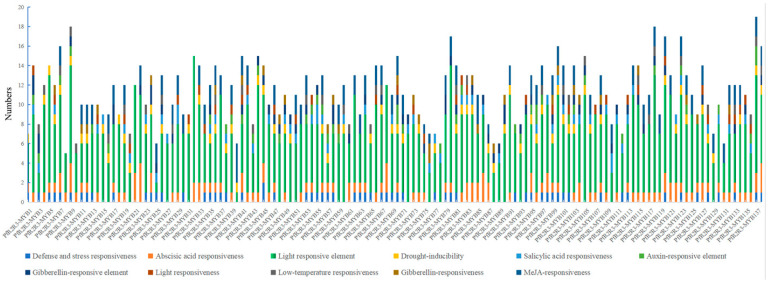
*Cis*-regulatory elements analysis of *PfR2R3-MYB* genes. Each *cis*-regulatory element type was marked by different colored boxes. Number of *cis*-regulatory elements in the promoter regions of *PfR2R3-MYB* genes.

**Figure 5 genes-14-00007-f005:**
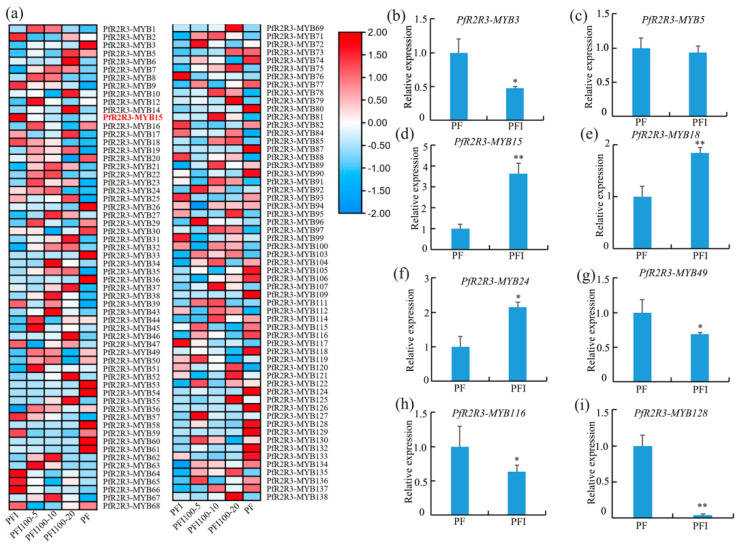
Expression pattern analysis of *PfR2R3-MYB* genes. (**a**) *PfR2R3-MYB* gene expression in phytoplasma-infected *P. fortunei* seedlings (PFI), PFI treated with 100 mg·L^−1^ Rif for 5, 10, and 20 days (PFI100-5, PFI100-10, and PFI100-20) and PF (*P. fortunei* seedlings). (**b**–**i**) QRT-PCR analysis of eight DEGs in PF and PFI. The error bars indicate the standard errors of three biological replicates. Asterisks indicate a significant difference (** *p* < 0.01; * *p* < 0.05) between PF and PFI.

**Figure 6 genes-14-00007-f006:**
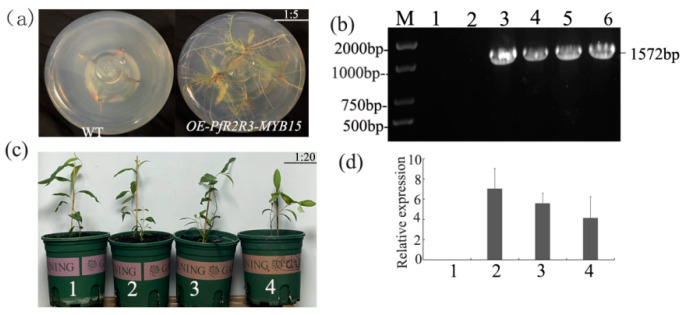
*OE-PfR2R3-MYB15* causes witches’ broom symptoms in *P. trichocarpa*. (**a**) The root morphology of the WT and the transgenic line. (**b**) PCR detection of *OE-PfR2R3-MYB15*. M: DNA marker, 1, WT (negative control); 2, H_2_O (negative control); 3, positive control (plasmid); 4, *OE-PfR2R3-MYB15-1*; 5, *OE-PfR2R3-MYB15-2;* 6, *OE-PfR2R3-MYB15-3*. (**c**) Phenotype identification. 1, WT, 2, *OE-PfR2R3-MYB15-1*; 3, *OE-PfR2R3-MYB15-2*; 4, *OE-PfR2R3-MYB15-3*. (**d**) QRT-PCR detection of *OE-PfR2R3-MYB15* transgenic plants. 1, WT; 2, *OE-PfR2R3-MYB15-1*; 3, *OE-PfR2R3-MYB15-2;* 4, *OE-PfR2R3-MYB15-3*.

**Figure 7 genes-14-00007-f007:**
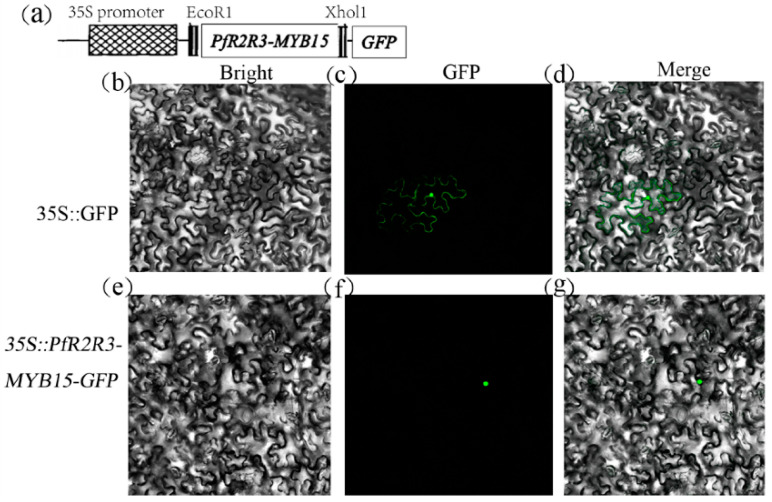
Subcellular localization of *PfR2R3-MYB15*. (**a**) Diagram of the 35S promoter and GFP in *PfR2R3-MYB15*. (**b**–**g**) 35S::*PfR2R3-MYB15-GFP* and 35S::GFP were transferred into *N. benthamiana*. Bright fields (**b**,**e**), GFP fluorescence detection (**c**,**f**), and superposition fields of GFP fluorescence (**d**,**g**) are shown.

**Figure 8 genes-14-00007-f008:**
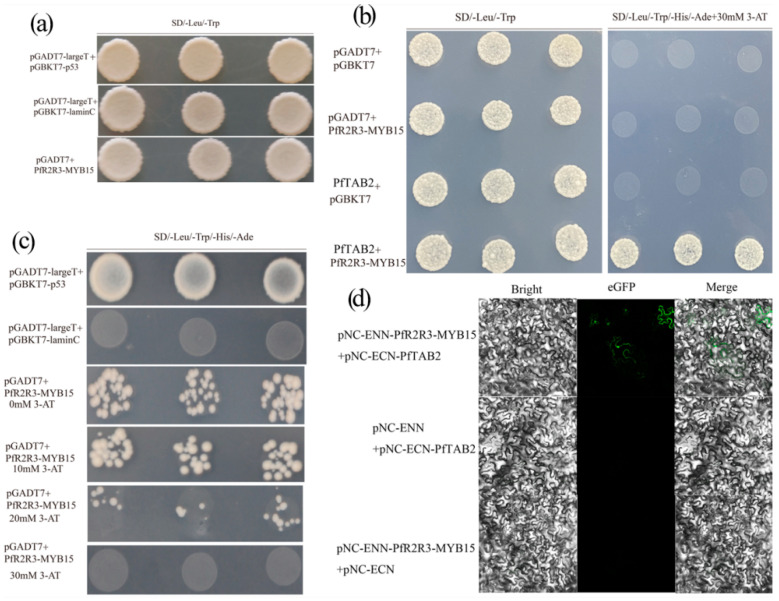
*PFR2R3-MYB15* interacted with *PfTAB2*. (**a**) pGADT7-largeT/pGBKT7-p53 were positive control, while pGADT7-largeT/pGBKT7-laminC were negative control. (**b**) 0, 10, 20, 30, and 40 mM 3-AT were used for *PfR2R3-MYB15* self-activating activity screening. (**c**) *PfR2R3-MYB15* interacted with Paulownia_LG3G001014 (*PfTAB2*) by Y2H screening. (**d**) *PfR2R3-MYB15* interacted with Paulownia_LG3G001014 (*PfTAB2*) by BiFC.

## Data Availability

All the sequencing data from *P. fortunei* transcriptome used in this study are available from the SRA-Archive (http://www.ncbi.nlm.nih.gov/sra accessed on 27 June 2021) of NCBI under the study accession numbers SRR11787883, SRR11787894, SRR11787905 and SRR11787912-SRR11787921.

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
