# Peer review of "Genome-Wide Analysis of Specific PfR2R3-MYB Genes Related to Paulownia Witches’ Broom"

_genes, 2022, doi:10.3390/genes14010007_

Round 1

Reviewer 1 Report

The author touched on the Pfr2r3-MYB gene analysis of the Paulownia fortune. The bioinformatics analyses are very impressive and clearly showed the Pfr2r3-MYB’s gene distribution in the whole genome. This research is very important for identifying the potential genes that are functional in preventing the Paulownia Witches’ broom. However, there are some errors in this manuscript that I would like to point out here. They are as follows:

1.     Title: a blank space should be inserted in front of ‘broom’.

2.     P4, L168; ‘All raw read dates’ should be ‘All the raw read data’.

3.     P5, L204; ‘The duplicated PfR2R3-MYB genes relationship’ should be ‘gene relationship’.

4.     P6, L213; why did the author construct the NJ tree between AtR2R3-MYB and PfR2R3-MYB genes, please? How were the gene numbers chosen, please? The related background should be added here.

5.     P6, L220; similar question, PfR2R3-MYB15 is clustered in the same subfamily XXIII with AtRAX2, so why it is important to work on PfR2R3-MYB15? Why All the other MYB proteins are not discussed here at all?

6.     P9, L244; ‘analysis’ or ‘analyses’ should be uniform in the whole manuscript. ‘gene’ or ‘genes’ should be uniform, too.

7.     P9, L255; the author mentioned ‘the related elements may be important for PfR2R3-MYB gene family’. I can’t understand why. The summary is confusing. Please explain.

8.     P9, L260; ‘PfR2R3-MYBs involved in the phytoplasma interaction and verification of qRT-PCR’ should be corrected and can’t be understood.

9.     P10, L285; ‘were’ should be ‘was’. Why was the branching phenotype of PfR2R3-MYB focused, please?

10.  P13, L406; the author mentioned ‘PfR2R3-MYB15 interacted with PfTAB2, which played vital roles in phytoplasma infection’. However, which kind of role is it, please? More discussion about this is needed, especially how PfTAB2-PfR2R3-MYB15 interaction functions in which step of Paulownia Witches’ broom.

In conclusion, this manuscript is well-designed and clearly exhibited the whole-genome-wide analyses of PfR2R3-MYB genes, which is interesting and very important. The only drawback is that English grammar needs to be double-checked throughout the whole manuscript.

Reviewer 2 Report

The manuscript by Zhao et al. conducted genome-wide characterization and expression profiling of Pfr2r3-MYB genes using RNA-seq and qRT-PCR analysis. Additionally, they conducted the functional characterization of PfR2R3-MYB15 using a transgenic approach, subcellular localization, yeast two-hybrid, and BiFC assay. After carefully reviewing the manuscript, the following shortcomings necessary to be resolved are detected:

1.       The authors created data for the Genes journal using only one OE-line. It is one of the major weaknesses in this manuscript. In order to publish in SCI/SCIE-indexed journals, the data from at least two transgenic lines should be provided. Thus, the authors must include the data from the second OE-line in this manuscript to publish in the Genes journal. Otherwise, remove this figure and all data related to this part from this manuscript because they are not reliable.

2.       Additionally, the authentication of the OE-line was conducted just by the PCR analysis. The gene expression levels of the overexpressed gene in the WT and the OE-lines must be analyzed.

3.       The quality of the figures in the manuscript is not good. They should be improved and adjusted to at least 300 dpi resolution.

4.       How do you certify the correctness of your phylogenetic tree? Which pipelines were used to construct the best phylogeny tree? Why was bootstrap analysis not conducted for the phylogenetic analysis?

5.       The data on Y2H assay is good and reliable. But, why was the localization result of BiFC assay and that of subcellular localization of PfR2R3-MYB15 not consistent? Because PfR2R3-MYB15 localizes solely to the nucleus, the localization result of BiFC assay should also be specific to the nucleus (if PfTAB2 does not localize to other compartments), not ubiquitous as in the figure 8d. Moreover, the information of BiFC assay (Figure 8d) is missing in the figure legend.

6.       I think it would be better if the motif numbers were labeled using TBtools software in figure 3b. In addition, information regarding the conserved motifs of R2R3-MYB proteins should be added as a supplementary figure. Please refer to the supplementary figure showing the conserved motifs in other papers.

7.       Make a new figure 4 by describing only the cis elements related to stress and hormones. The colors of each cis element should contrast with each other. The old figure 4 can be added as a supplementary figure to the manuscript. The supplementary table of all the cis-regulatory elements identified in this gene family should be added to the manuscript.

8.       Make new bar graphs for figure b-i using 2−ΔΔCT method to show the fold change differences. For the fold-change values of the analyzed genes, please describe them in the text. If the fold change is greater than 2, it is significant. So, the genes whose fold change is greater than 2 could be described as the significant genes that you should emphasize in your studies.

9.       A scale bar should be added in figures 6a and c. Figure 6a should be replaced with a figure showing the root morphology of the WT and the transgenic line. Please refer to other papers to make this figure. Figure 6c should show the pots without cropping them untidily. Please refer to other good papers to make this figure.

10.   Although Jennifer Smith, PhD, from Liwen Bianji, Edanz Group China 427 (www.liwenbianji.cn/ac) is acknowledged for English editing of a draft of this manuscript, the submitted version of this manuscript contained a lot of errors and a lot of parts, including the title, in the manuscript should be significantly edited and improved. I suggested the manuscript should be edited by language editing services specializing in plant biology, such as Plant editors.

11.   Rather than mentioning “using the TBtools (V1.049)”, please describe which algorithm or program in the TBtools (V1.049) was used for each in silico analysis. I suggested the authors read other good papers carefully and revise the manuscript intensively to significantly improve the presentation quality of their manuscript.

12.   Line 126, “Expression levels were calculated by the 2−ΔΔCT method [30]”. I do not think so. If 2−ΔΔCT method is used for the analysis, the relative expression values of the control samples should be 1. Please, read other good papers and discuss with the senior if you do not understand my comment.

13.   Line 382-383, “so we analyzed the expression patterns of the PfR2R3- 382 MYB genes in PF and PFI by RNA-Seq.” The RNA-Seq data used for the analysis should be described in details, such as from which database they were downloaded, how the raw RNA-seq data were processed using which tools, etc., in the materials and methods section.

14.   For the gene expression analysis via qRT-PCR, how plant materials were prepared, how they were treated with the pathogen, and how the samples were collected for the analysis should be described in the materials and methods section. The quality of the presentation was not good in this submitted version of the manuscript and should be significantly improved.

15.   Do not upload supplementary materials twice by uploading them in the non-published material section as well.

16.   The caption of supplemental table 1 should be revised with the help of an English language expert. The PF and PFI could be clarified in the table footnote.

17.   The caption of supplemental table 5 should be revised. Please refer to other papers for the appropriate table caption.

18.   The caption of supplemental table 6 should be revised. Please refer to other papers for the appropriate table caption. Please compute the value of million-years ago of the duplicated gene pairs and add the type of selection for all duplicated gene pairs. Please refer to other papers for the preparation of this table.

19.   The caption of supplemental table 7 should be revised with the assistance of the language expert. In addition, other data related to PF and PFI (i.e., their expression levels, p-values or FDR, etc.) should be added to the table so that we can know the values of log2FC in the table are reliable.

20.   Almost all of the figure legends and table captions in the manuscript should be re-edited and significantly improved.

21.   Many issues and weaknesses have been investigated in this manuscript. They should be carefully revised before publication. The quality of presentation and many data in the manuscript are not good and strong enough for publication in Genes and should be intensively revised.

Reviewer 3 Report

Gene Review

The manuscript entitled “Genome-Wide Analysis of Specific Pfr2r3-MYB Genes Related 2 to Paulownia Witches’ broom” by Zhao et al., have done comprehensive analysis of Myb Transcription factors and identified 138 R2R3 Myb’s in Paulownia an economically important tree species. Expression analysis of identified Mybs in response to phytoplasma infection showed that PfR2R3-MYB15 was highly upregulated. Additionally, overexpression of PfR2R3-MYB15 in Populus showed increased branching phenotype in transgenic plants. Although, author have done sufficient work for characterization, some points need to be addressed before it is accepted for publication.

Major point:

Majority of result section is very poorly written. However, characterization part under heading 3.7 and 3.9 should be rewrite.

Minor points:

1.       It would be better if author have included poplar sequence also for phylogenetic analysis.

2.       Line 254- Provide the names of common cis acting element.

3.       Is there any specific reason to conduct expression analysis in response to phytoplasma infection?

4.       Explain PFI, PFI100-5, PFI100-10, PFI100-20 and PF in result and Figure legend.

5.       In Figure 5a, in heatmap expression of PfR2R3-MYB15 seems higher than PfR2R3-MYB18. However, in qRT-PCR data values looks opposite. It looks like expression of PfR2R3-MYB18 is almost double than PfR2R3-MYB15 in PFI condition and the difference is significantly higher for PfR2R3-MYB18 under PFI condition as compared to control condition. Explain.

6.       Include positive control in PCR (Figure 6B). Mention target size as well in Figure 6b.

7.       Figure 6c. Are these different over expression lines or different plants of single line? Write the age of the plants used for branching phenotype.

8.       Figure 8 legend are mislabeled.

9.       In BiFC, what is site of interaction? As PfR2R3-MYB15 is nuclear localized and location of Arabidopsis ATAB2 is chloroplast, what could be the potential reason for their interaction? Explain.

Round 2

Reviewer 2 Report

My comments can be found in the attached files as a sticky note and comments in the Microsoft word document.

Reviewer 3 Report

The authors have responded very well to all the queries. 

Author Response

We would like to thank you for giving us constructive suggestions which would help us in depth to improve the quality of our manuscript.